# An Investigation on the Possible Application Areas of Low-Cost PM Sensors for Air Quality Monitoring

**DOI:** 10.3390/s23083976

**Published:** 2023-04-14

**Authors:** Domenico Suriano, Mario Prato

**Affiliations:** ENEA—Italian National Agency for New Technologies, Energy and Sustainable Economic Development, Department for Sustainability—Brindisi Research Center, SS. 7 Appia, km 706, 72100 Brindisi, Italy

**Keywords:** air quality monitoring, low-cost sensors, EPA guidelines, PM sensors, sensor evaluation, field evaluation, air pollutants, gravimetric method

## Abstract

In recent years, the availability on the market of low-cost sensors (LCSs) and low-cost monitors (LCMs) for air quality monitoring has attracted the interest of scientists, communities, and professionals. Although the scientific community has raised concerns about their data quality, they are still considered a possible alternative to regulatory monitoring stations due to their cheapness, compactness, and lack of maintenance costs. Several studies have performed independent evaluations to investigate their performance, but a comparison of the results is difficult due to the different test conditions and metrics adopted. The U.S. Environmental Protection Agency (EPA) tried to provide a tool for assessing the possible uses of LCSs or LCMs by publishing guidelines to assign suitable application areas for each of them on the basis of the mean normalized bias (MNB) and coefficient of variance (CV) indicators. Until today, very few studies have analyzed LCS performance by referring to the EPA guidelines. This research aimed to understand the performance and the possible application areas of two PM sensor models (PMS5003 and SPS30) on the basis of the EPA guidelines. We computed the R^2^, RMSE, MAE, MNB, CV, and other performance indicators and found that the coefficient of determination (R^2^) ranged from 0.55 to 0.61, while the root mean squared error (RMSE) ranged from 11.02 µg/m^3^ to 12.09 µg/m^3^. Moreover, the application of a correction factor to include the humidity effect produced an improvement in the performance of the PMS5003 sensor models. We also found that, based on the MNB and CV values, the EPA guidelines assigned the SPS30 sensors to the “informal information about the presence of the pollutant” application area (Tier I), while PMS5003 sensors were assigned to the “supplemental monitoring of regulatory networks” area (Tier III). Although the usefulness of the EPA guidelines is acknowledged, it appears that improvements are necessary to increase their effectiveness.

## 1. Introduction

Many studies have proved the existence of a direct link between exposure to air pollutants and issues concerning public health or climate change [1,2,3,4,5]. Air quality monitoring is controlled by national regulations, and the equipment required to meet the standards established by such regulations is characterized by high costs due to purchasing, maintenance, and logistical issues [6,7,8,9,10,11]. For this reason, in many cases, fixed monitoring station networks of governmental agencies feature few nodes that are sparsely deployed across the territory. As a consequence, it is not often possible to obtain pollutant maps with an adequate spatio-temporal resolution [9,12].

In recent years, an appealing solution to this issue has been represented by the rising of air quality monitors based on low-cost sensors [10,11,12]. A remarkable number of research institutions and companies have started to design, produce, and test a huge variety of sensors not only for pollutant gas and particulate matter monitoring [10,11,13], but also for malodor detection [14,15]. The use of low-cost sensors (LCSs) or low-cost monitors (LCMs) based on LCSs for air quality monitoring has been investigated and explored by several studies reporting interesting potentialities, but also substantial limitations and caveats [9,10,11,12,13,16,17,18,19].

The technologies featuring LCSs and LCMs provide devices that are ten or more times cheaper than the regulatory instrumentation [9,11], but their data quality is questionable [9,11,17,19]; additionally, the performance information provided by the manufacturers of LCMs/LCSs is limited in most cases. Several studies have already addressed this issue, and various strategies have been explored to improve LCS or LCM performance. These range from the employment of sensor arrays [20] to the use of various data elaboration algorithms, such as multilinear regression or artificial neural networks [9,11,20,21]. The process of improving the performance of these devices by post-processing their data is commonly termed calibration.

Several studies suggest that it is preferable to evaluate or calibrate the performance of such devices in the environment of their final deployment (more concisely, “on-field”), which could be an outdoor site or an indoor space [9,11,13,16,18,19,21]. The on-field evaluation or calibration of LCSs or LCMs is performed by co-locating the device under test with reference instrumentation featuring higher standards of accuracy and precision [11].

The performance of the devices under test can be assessed through indicators calculated utilizing the data provided by the reference instrumentation and the data of the devices under evaluation or calibration. We found that the most commonly used indicators in studies concerning LCS/LCM evaluation or calibration were the coefficient of determination (R^2^), the root mean squared error (RMSE), the mean absolute error (MAE), the mean normalized bias (MNB), and the coefficient of variation (CV). The R^2^ indicator describes how well the LCM/LCS correlates with the reference device; it ranges from 0 to 1. Values close to 0 indicate poor performance, while values close to 1 show a good agreement between the device under test and the reference device. RMSE, MAE, and MNB are indicators related to the extent of the error between the measurements of the LCM/LCS and the measurements of the reference device; values close to 0 represent a good performance. The coefficient of variation (CV) is used to describe the extent of the variation displayed in the measurements provided by several samples of the same LCM/LCS model under evaluation; values close to 0 indicate a good level of consistency for the model.

The study described in this manuscript focused on the evaluation of the performance of two LCS models designed for measuring particulate matter (PM) concentrations and their application areas. PM is an air pollutant composed of microscopic particles whose aerodynamic diameter is less than or equal to 10 µm, in the case of PM_10_, or less than or equal to 2.5 µm or 1 µm, in the case of PM_2.5_ and PM_1_, respectively.

The present manuscript is organized as follows. In Section 2, an overview of previous related works is presented together with an overall description of this study. Section 3 provides some necessary background information that is useful to fully understand some fundamental aspects of this research. The materials and methods used to perform this study are reported in Section 4, while the results are shown in Section 5. A detailed discussion of the results can be found in Section 6. The findings presented in this penultimate section led to the conclusions summarized in Section 7 of this article.

## 2. Related Works and Study Description

Although in recent years, several studies concerning the evaluation/calibration of PM sensors already available on the market have been conducted, it is quite difficult to compare their results due to the remarkable heterogeneity of conditions under which they were performed. By reading the scientific literature, it was found that they differed in terms of the test environment (outdoor, indoor, or in a laboratory test chamber), reference instrumentation used, performance indicators, dataset structures (e.g., data grouped by hourly or daily means), and test duration. All these aforementioned factors directly affected the quantification of the performance, and this element was the origin of the difficulties in comparing the results, even though we considered studies that used identical indicators.

Gao et. al. [22] evaluated the performance of a Shinyei sensor measuring PM_2.5_ during a 4-day test performed in an outdoor environment, concluding that it correlated better with optical reference instruments (R^2^ = 0.86–0.89) than with gravimetric ones (R^2^ = 0.53). Vogt, Castell, et al. [23] carried out a performance evaluation of the PMS5003, SPS30, and OPC-N3 sensors through an outdoor test that lasted 7 weeks by using optical and gravimetric reference instruments. They found that in the case of PM_2.5_, the sensors showed a good performance in terms of the coefficient of determination (R^2^ = 0.7–0.9), while they performed worse for PM_10_ measurements (R^2^ = 0.6–0.7). Kosmopoulos, Kazantzidis, et al. [24] evaluated and calibrated a PMS5003 sensor integrated into an LCM called PurpleAir-PA II by performing an experiment that lasted 18 months. In this experiment, the authors gathered data representing hourly measurements provided by the LCM and an optical counter as a reference, reporting R^2^ = 0.81 for PM_1_, R^2^ = 0.56 for PM_2.5_, and R^2^ < 0.37 for PM_10_. Masic et al. [25] evaluated the PMS5003 and OPC-N2 sensors through a test carried out in a heavily polluted outdoor environment. They showed that by considering daily averaged measurements, the sensors correlated well with the reference (R^2^ = 0.9–0.95), even though the MAE ranged between 29.4 µg/m^3^ and 55.2 µg/m^3^.

Other works [26,27,28] have explored the potentialities and limits of various PM sensor models by applying them in different scenarios.

Evaluations of a notable variety of LCSs/LCMs for PM concentration measurements can be found in the AQ-SPEC program [18]. This study has remarkable importance not only for the number of different LCM/LCS models investigated, but also because they were all evaluated under the same conditions. In this work, the devices under test were compared on-field or in the laboratory with different types of reference device: beta attenuation monitors and optical counters. The duration of the evaluation period was fixed at roughly two months for every device tested, and the data gathered were grouped into 5 min, hourly, and daily means. The indicators used for the evaluations were the coefficient of determination (R^2^) and the coefficient of variation (CV).

The study presented in this manuscript was performed under similar conditions: two copies of two different PM sensor models (PMS5003, produced by Plantower, and SPS30, produced by Sensirion [29,30]) were evaluated on-field through a test that lasted roughly two months. The purpose of our study was to understand the potential uses, or application areas, of these PM sensors presenting a good price–quality ratio. To accomplish this task, we took as a reference the Williams et al. [19] report published by the U.S. Environmental Protection Agency (EPA). In this report, the potential uses of LCSs were classified on the basis of certain performance indicators, more precisely, the MNB and CV [13,19]. In particular, we investigated if the PM sensors could be used for supplemental monitoring to complement the existing fixed PM stations of the local government environmental protection agency named ARPA Puglia [31]. As indicated in the EPA report [19], if the sensor performance presents a variance and a bias error of less than 20% (CV and MNB) and a data completeness greater than 80%, the devices can be reliably used as a supplemental monitoring tool for improving the spatial resolution of the pollutant concentration maps produced by ARPA Puglia [32]. As seen in previous works [18,23], the reference instrumentation used for evaluating the LCSs/LCMs directly affects the indicator performance; moreover, SPS30 and PMS5003 have already been evaluated in the AQ-SPEC program [18] using beta attenuation monitors and optical counters as a reference. Considering these factors, we judged it important to use the gravimetric measurements provided by ARPA Puglia in order to understand the possibility of employing these PM sensors to complement ARPA Puglia data. These considerations, in addition to the fact that the MNB of the sensors was not calculated in the AQ-SPEC project [18], led to the design of the experiment hereafter described.

## 3. Background

As mentioned earlier, the conditions characterizing the assessment or the calibration of the LCSs/LCMs are very important for their final evaluation. The test environment is the most influential factor. In general, LCMs or LCSs evaluated or calibrated in a laboratory test chamber show better performance in comparison with devices evaluated or calibrated on-field. This is because the environmental variables typical of real-world scenarios, which negatively affect LCM/LCS performance, are hard to reproduce in a laboratory test chamber. Other relevant factors weighing on their performance quantification are the type of data grouping, the instrumentation used as a reference, the duration of the evaluation period, the range of the pollutant concentrations, and the performance indicators adopted. As already mentioned, there is no commonly accepted procedure used to perform LCM/LCS evaluations or calibrations. In this respect, the EPA guidelines tried to standardize these processes by proposing a minimum set of rules to allow a better comparison of LCS/LCM performance. In particular, they proposed the adoption of the CV and MNB indicators to provide reliable information about the optimal use of LCSs/LCMs.

Therefore, in order to assess the possible uses of the sensors following the approach indicated in the EPA report [19], the MNB and CV indicators were calculated for each sensor model. In addition to these, the coefficient of determination (R^2^), RMSE, and MAE were also computed for each copy of the sensors to allow a comparison with previous studies where the same sensors were evaluated or calibrated. R^2^, MAE, MNB, RMSE, and CV are performance indicators defined as follows (see also [9,13,21,23,33]):

(1)$$R^{2}=\frac{{\left(\sum_{i=1}^{N}\left(s_{i}-\bar{s}\right)(r_{i}-\overset{-}{r})\right)}^{2}}{\sum_{i=1}^{N}{\left(s_{i}-\bar{s}\right)}^{2}{\left(r_{i}-\overset{-}{r}\right)}^{2}}$$(2)$$MAE=\frac{1}{N}\sum_{i=1}^{N}\left|s_{i}-r_{i}\right|$$(3)$$MNB=\frac{1}{M}\sum_{j=1}^{M}\frac{1}{N}\sum_{i=1}^{N}\frac{{(s}_{i,j}-r_{i})}{r_{i}}$$(4)$$RMSE=\sqrt{\frac{1}{N}\sum_{i=1}^{N}{\left(s_{i}-r_{i}\right)}^{2}}$$(5)$$CV=\frac{\sqrt{\frac{1}{MN-1}\sum_{j=1}^{M}\sum_{i=1}^{N}{\left(s_{i,j}-s_{i}\right)}^{2}}}{\frac{1}{M}\sum_{j=1}^{M}\frac{1}{N}\sum_{i=1}^{N}s_{i,j}}$$
where $r_{i}$ is the *i*th measurement of the reference, $s_{i}$ is the *i*th reading of the sensor, N is the total number of observations, $\bar{s}$ is the average of the sensor readings, and $\bar{r}$ is the average of the reference measurements. Concerning the CV and MNB formulas, $s_{i,j}$ represents the *i*th reading of the sensor of the *j*th copy of the sensor model, while *M* is the number of copies for each sensor model, which was equal to two in our case.

The classification of the LCS/LCM uses, or application areas, proposed by the EPA features five tiers (Tier I–Tier V), presented in Table 1, which depend on the pollutant considered, the MNB and CV values.

Once the CV and MNB have been computed for a sensor model, it is possible to determine the optimal “tier”, or application area, of the LCM/LCS. The conditions necessary to assign an LCS/LCM model to a tier are determined by the “AND” logic operation for the ranges of the MNB and CV values indicated in Table 1. Thus, as an example, an LCM/LCS model belonging to Tier IV must demonstrate both −0.3 < MNB < 0.3 and CV < 0.3.

## 4. Materials and Methods

The LCSs available on the market for air quality monitoring are devices able to measure pollutant concentrations using different technologies and working principles. Nonetheless, aside from the sensor type and technology used, these devices need appropriate electronic boards for their effective use. The electronic circuitry of these boards converts the current or the voltage output of the sensing element, dependent on the pollutant concentration, into an electronic signal available at the output interface. These sensors can present various types of interfaces, the most common being: analog, TTL serial, I2C, and USB. To effectively make use of the data produced by the sensors, it is required a suitable electronic system capable of reading the signals coming out of the output interfaces and converting them into usable data. The overall electronic system in charge of accomplishing this task is commonly called a low-cost monitor (LCM).

The PM sensors evaluated in this work were integrated into the SentinAir platform [34] designed to act as both a tool for quickly performing evaluations/calibrations of LCSs and an LCM for indoor or outdoor environments [21,35,36,37]. The SentinAir system is an in-house and open-source design implemented in the ENEA research center of Brindisi, located in the Puglia region of Italy. Therefore, all the materials, software, and procedures required to build a copy of SentinAir are available online on the project repository webpage [34].

The main difference between the SentinAir system and the other commercially available LCMs is represented by the possibility of integrating, or installing, a huge variety of sensors presenting any of the earlier mentioned output interfaces and produced by various manufacturers. This capability was achieved thanks to the adoption of the low-cost micro-computer Raspberry 3B+ [38] as the core of the system and the software created for the operation of SentinAir. As a matter of fact, the Raspberry 3B+ hardware is characterized by four USB ports, an I2C bus, and a TTL serial port, while a driver software for the installation of an analog-to-digital converter (ADC) was designed and implemented to allow the use of the ADCPi board by ABelectronics [39]. This electronic board is necessary for the use of the LCSs with an analog output interface. A software system composed of drivers written in the Python language [40] specific to each device or sensor installable in SentinAir provides the “plug-and-play” feature for the system. Another very useful feature of SentinAir consists in its ability to be remotely operated through its dual wireless communication system: a WiFi channel and an internet connection via a USB modem.

The SentinAir device used for the test acted as an LCM containing two copies of the PMS5003 sensor model and two copies of the SPS30 model, as illustrated in Figure 1. The LCSs used in this experiment were particle optical counters available on the market, whose hardware was contained in a compact case, as shown in Figure 2.

Their working principle is illustrated in Figure 3. It is based on a laser beam scattered by the particles entering a detection camera. The more particles cross the laser beam, the more scattered the beam becomes. The beam is detected by a light detector that provides an electronic signal depending on the scattering level of the laser light. Thus, the microprocessor inside the sensor is devoted to translating the electronic signal of the detector into numbers of particles per volume unit. Therefore, an algorithm implemented in the sensor microprocessor provides the PM concentration depending on the number of particles detected. Unfortunately, the manufacturers do not provide details about this algorithm. Concerning the output interface, these LCSs are provided with a serial TTL interface in the case of the PMS5003 sensor, and an I2C or a serial TTL interface in the case of the SPS30 sensor. By connecting the LCM hardware through these interfaces, measurements related to PM_10_, PM_2.5_, and PM_1_ (and, in the case of SPS30, also PM_4_) concentrations can be read. More details about these LCSs can be found by downloading the datasheets from the manufacturer’s websites [29,30].

Other LCMs similar to SentinAir are currently available on the market. These are all based on LCSs whose working principle follows the simplified scheme depicted in Figure 3. An estimation of their costs is provided in Table 2, where some examples of both LCMs and LCSs for PM measurements are listed.

The experiment designed to assess the potential uses of these sensors consisted in placing a SentinAir device near the ARPA Puglia fixed monitoring station located in the town of Mesagne (Italy), as shown in Figure 4.

The reference used for the evaluation was the ARPA Puglia monitoring station located in Via Udine, which is devoted to measuring the background concentrations of PM_10_ in a suburban environment. It performs the measurements using gravimetric instruments and provides data concerning the daily means of this pollutant that are freely downloadable from the ARPA website [32].

Data elaboration and indicator computations were performed using the Scikit-learn libraries written in the Python language [41,42,43], which are an open-source software freely downloadable from their website [41]. The sensor data were read by the SentinAir device every 5 min, and, subsequently, daily means were computed for building the database together with the ARPA data. Italian rules for PM monitoring require just the monitoring of PM_10_ on a daily average basis and yearly averages for PM_2.5_; for this reason, the useful data provided by the fixed reference station were only those related to the daily averages of PM_10_. The SentinAir system can store the data related to the performed measurements on its internal SD card memory; moreover, it features a web server from which the user can download any data measured by the device thanks to the presence of a USB modem [35,36,37]. The data gathered by the SentinAir device were periodically downloaded and joined with the ARPA measurements that are publicly available on the ARPA website [32]. Starting from the daily averages of the ARPA monitoring station and the daily averages computed by the LCM, it was possible to assess the LCS uses related to PM_10_ monitoring by considering the MNB and CV values computed for each sensor model according to the EPA guidelines [13,19].

The performance of PM sensors is negatively affected by the environmental relative humidity [44,45]. The reason for this effect is the condensation of water vapor that makes the aerosol particles comprising the particulate matter grow hygroscopically. This phenomenon causes incorrect measurements in devices based on the optical counter working principle, such as the PM sensors considered in this experiment. Some studies [44,45] have proposed an algorithm to take into account the negative effect of humidity in order to improve sensor performance. This algorithm consists in applying a correction factor to the sensor readings, as indicated by the following formulas:(6)$${PM}_{corrected}=\frac{{PM}_{uncorrected}}{C}$$
(7)$$C=1+\frac{\frac{k}{1.65}}{\frac{100}{RH}-1}$$
where *RH* is the relative humidity, and “*k*” is a parameter depending on the nature of the particulate matter. To have a complete view of the capabilities of the sensors considered in our investigation, we applied the correction algorithm illustrated above to understand the extent to which the performance of the sensors could be improved. However, in our experiment, no information could be collected on the composition of *PM* compounds; therefore, we selected two distinct values (*k* = 0.5 and *k* = 0.62), as suggested in the works of Crilley [44] and Di Antonio [45]. In the first study, it was stated that the expected range of the “*k*” parameter could be reasonably thought to be 0.48–0.51 for PM_10_, while the second study hypothesized that the “*k*” value could be set equal to 0.62 in the case of a mixture of organic and inorganic compounds, such as in a typical polluted urban environment.

## 5. Results

For the purpose of this study, measurements from the 15th of September 2022 to the 27th of November 2022 were carried out. These data formed a dataset composed of the daily averages of PM_10_ concentrations resulting from the measurements of the four LCSs involved in the study (hereafter named “PMS5003(1)”, “PMS5003(2)”, “SPS30(1)”, and “SPS30(2)”) and the fixed reference station of ARPA Puglia (hereafter named “reference”). In the period indicated above, ARPA Puglia did not provide data for ten dates due to the maintenance of the instruments or other unknown issues, while the data recovery percentage was close to 100% for each sensor. For this reason, we excluded the records of the database lacking reference measurements from the computation of the performance indicators.

### 5.1. Results of the Performance without Considering the Humidity Effect

One of the aims of this study was the characterization of “out-of-box” PM data offered by the four sensors available on the market; thus, in this first stage, we assumed that the factory calibrations performed by the manufacturers would reflect the PM concentrations in the best way.

In Figure 5, the time series related to the LCSs under evaluation are reported along with the measurements of the reference. In this figure, it can be noted that higher PM_10_ concentrations were more frequent in the latter period of the experiment. We could explain this element by considering that, as the colder days approached, the use of wood burners, which are widely employed in the town for domestic heating, became more frequent. By the examination of this figure, it can also be noted that, in general, the SPS30 model tended to underestimate the PM_10_ concentrations, while the PSM5003 model tended to overestimate them. Nevertheless, we observed substantial agreement between the measurements of the two copies of the sensors for each of the two models. Another finding shown in Figure 5 was that the PM_10_ concentrations measured by the reference ranged from 7 µg/m^3^ to 51 µg/m^3^. Concerning the on-field evaluation of LCSs, wider ranges of pollutant concentrations increase the probability of reporting a better LCS performance in terms of R^2^. In this regard, the maximum value registered during the experiment was barely higher than the limit fixed by the Italian regulations, which is 50 µg/m^3^. This value was reported on only one day, the 26th of November 2022. The average PM_10_ concentration measured by the reference was equal to 20.4 µg/m^3^, less than half the limit fixed by the Italian regulations.

As for the consistency, or variability, both the models showed a good performance, and, as also underlined by the time-series plots, substantial agreement between the two copies of the same models can be observed in Table 3, which highlights the high correlation between them. As a matter of fact, all the values reported in this table are very close to 1, indicating very good intra-model consistency.

The ability of each sensor to reflect the reference measurements could also be evaluated by examining the plots presented in Figure 6. They show that the PMS5003 model correlated slightly better with the reference device than the SPS30 sensors. As a matter of fact, the R^2^ was 0.61 for both copies of the PMS5003 model, while the SPS30 sensors had an R^2^ equal to 0.57 and 0.55. The slope of the linear fit line computed for the two models explained both the tendency of the PMS5003 model to overestimate the PM_10_ concentrations and the fact that the SPS30 model tended to underestimate them. The plots in Figure 6 show that the slope of the PMS5003 model was roughly threefold the slope of the SPS30 sensors; however, the most important aspect is represented by the position of the linear fit line, which lies over the 1:1 reference line in the case of the PMS5003 model, and underneath it in the case of the SPS30 model. It is useful to recall that the slopes of linear fit lines close to unity and bias values near 0 denote the good performance of the device under evaluation. From Figure 6, it can also be noted that the values related to the slope and bias were very similar for the two copies of the same models. This finding, in conjunction with the values reported for the intra-model correlation, confirmed the consistency of both LCS models.

The complete overview of the sensor performance is provided in Table 4, where the values of the R^2^, MAE, RMSE, MNB, and CV indicators are presented. Concerning R^2^, MAE, and RMSE, the performance of the two models was quite similar, while a substantial difference was observed only in the MNB values. A possible explanation for this difference can be found by analyzing Figure 6. In the plots of this figure, one can note that both the linear fit lines of the PMS5003 model are very close to the 1:1 reference line, or rather, there is an intersection around the 10 µg/m^3^ concentration level. On the contrary, in the case of the SPS30 model, the fit lines lie underneath the reference line in a more distant position.

### 5.2. Results after Applying the Correction Factor for the Humidity Effects

As explained earlier, an algorithm to correct the negative effect of humidity was applied to the sensor measurements to evaluate potential improvements in their performance. For this reason, the relative humidity (RH) was measured and logged along with the other sensor data to compose the final dataset. Figure 7 shows the levels of relative humidity registered during the period of the experiment.

The RH values were used to compute the corrected measurements of the sensors as indicated in Equations (6) and (7). Once we had performed the correction, the daily averages were subsequently calculated for comparison with the reference data. From Figure 7, it can be noted that the range of RH values was between 22% and 88%. This finding gave us an idea of the variability in the conditions during the experimental period in terms of humidity levels, while the average was equal to 62%.

Table 5 summarizes the performance indicators of the sensors considering the humidity effect and compares them with their uncorrected measurements to quantify the effectiveness of the correction algorithm.

The data summarized in the above table reflect the trends of the sensor measurements, which are more extensively illustrated in Figure 8.

The analysis of the data presented in Table 5 showed that the corrective formulas to include the humidity effects produced an improvement in the performance of the PMS5003 sensor model. This finding was corroborated by the increase of R^2^ values, and the decrease of MAE and RMSE values with respect to the case of the uncorrected sensor measurements (denoted in the table as PMS5003(2) and PMS5003(1)). This phenomenon characterized both copies of the PMS5003 model (PMS5003(2) and PMS5003(1)), even though the PMS5003(2) model corrected using the 0.5 value for the “k” parameter showed a slightly better performance compared with the other cases.

Table 5 also reports that, in the case of the SPS30 model, the corrective algorithm could not provide an improvement for both copies of the sensor. The decrease in R^2^ values, the increase in MAE and RMSE, and the widening of the gap between the slope parameter and the unity value indicated the worsening of their performance.

To understand the reasons for the different outcomes after applying the correction to the two sensor models under evaluation, it is necessary to consider that the effect of the humidity was the hygroscopic growth of the aerosol particles due to the condensation of water vapor. This phenomenon led to an overestimation of the PM concentration by the sensors, which could not distinguish the real dimensions of the particles. To counterbalance this effect, the concentration read by the sensor was divided by the “C” factor (see Equation (6)), which is linked to the ambient relative humidity: the greater the RH value, the higher the value of the “C” factor is. As can be seen in Figure 5, Figure 6 and Figure 8, the uncorrected measurements of the PMS5003 sensors tended to overestimate the real PM concentration provided by the reference; thus, the application of the Equations (6) and (7) decreased the values of the readings performed by these sensors, making them closer to the real PM concentrations. On the contrary, the uncorrected measurements of the SPS30 model underestimated the real PM concentrations (see Figure 5, Figure 6 and Figure 8); therefore, the application of the correction factor contributed to further lowering the value of the sensor readings, causing a widening of the gap between the real concentration levels and those provided by the sensors.

## 6. Discussion

### 6.1. Analysis of Results

The results concerning the performance of the models evaluated in this study can be better understood by comparing them with the data found in previous, similar works. We mentioned earlier that a comparison with several existing studies was not an easy task due to the different conditions surrounding the various experiments and the different metrics used for assessing the LCM/LCS performance. Nevertheless, we selected two such studies with very similar characteristics to the present experiment.

In their research, Vogt and Castell [23] performed an on-field evaluation of three LCM models. Two of the devices under evaluation were the Ensense and the Airly LCM model, commercially available at the current date. The operation of the Ensense model is based on the PMS5003 sensor, while the Airly monitor uses an SPS30 sensor. The duration of their research (53 days) was shorter than the one of the experiment described in the present study, but similar to it. They assessed the performance of the LCMs using gravimetric and optical reference instrumentations and by considering the hourly and daily averages of the measurements. In Table 6, the results related to the sensors involved in the present study are compared with the results from Vogt and Castell’s study by considering only the daily averages.

In the same table, the performance indicators determined by the AQ-SPEC program [18] are presented for comparison with this research. In this work, a notable number of LCMs and LCSs were evaluated on-field using both optical and beta-attenuator reference instrumentations. The assessment of the devices was performed by considering 5 min, hourly, and daily averages; however, in accordance with the present study, only the results related to the daily averages are summarized. Another element in common with the present work characterizing the AQ-SPEC program was the duration of the LCS/LCM tests, which was roughly equal to two months for each device under evaluation.

By inspecting the table above, one can note a marked heterogeneity in the values of almost all the indicators related to the same sensor model. The reasons for these wide ranges are ascribed to various factors, mainly, different reference types, and PM concentration ranges used in the evaluations. The table also shows that data completeness could be found only for the R^2^, slope, and bias parameters.

If we consider the LCMs using the SPS30 sensor separately from those employing the PMS5003 model, we can see that the ranges of R^2^ were 0.18–0.9, and 0.06–0.89, respectively. Therefore, the values found in this work for PMS5003 (R^2^ = 0.61) and SPS30 (R^2^ = 0.55–0.57) indicate better performance in relation to the median value of R^2^ found by the previous studies (R^2^ = 0.41 for PMS5003 and R^2^ = 0.36 for SPS30).

Table 6 also indicates that the ranges of the RMSE for SPS30 and PMS5003 were, respectively, 1.92–9.47 µg/m^3^ and 3.79–35.3 µg/m^3^. If we compare these ranges with the data found in this experiment (RMSE = 9.19–9.47 µg/m^3^ for SPS30, RMSE = 11.63–12.09 µg/m^3^ for PMS5003), we can note that the SPS30 displayed a worse performance, even though it has to be said that there was just one record in the table to compare it with; meanwhile, in the case of the other sensor model, the RMSE values were placed in the middle of the range found by the previous studies.

The other indicator presenting a completeness of data was the slope of the linear fit (see also Figure 6). As shown in Table 6, the ranges were equal to 0.26–1.61 for the SPS30 model and 0.64–3.38 for the PMS5003 model. In this regard, by considering that the ranges found in this study were, respectively, 0.59–0.60 and 1.38–1.41 for the SPS30 and PMS5003 models, we could conclude that the SPS30 model evaluated in this study showed lower slopes with respect to the overall median value (equal to 0.93), while the PMS5003 model was characterized by slopes more in line with the overall median value (equal to 1.37).

The coefficient of variation (CV) is the indicator that provides quantitative information about the consistency of the sensors or devices belonging to the same model. The values found by this work (1.96% for PMS5003 and 2.76% for SPS30) were much lower than the maximum CV found by the previous study (equal to 5.8% for PMS5003 and 11% for SPS30).

Unfortunately, we could not find MAE values for the SPS30 sensor in any previous works, but in the case of the PMS5003 model, they ranged from 9.3 µg/m^3^ to 32.6 µg/m^3^. Thus, the PMS5003 model evaluated in this study showed a better performance in terms of MAE values.

As shown in Table 1 and Table 4, the PMS5003 model featured an MNB of 0.14 and a CV of 1.96%, which fall in the range of the values characterizing Tier III. Thus, according to the classification proposed by the EPA guidelines, this sensor model can be used for supplemental monitoring to complete the data provided by the regulatory monitoring network and achieve a better spatio-temporal resolution for pollutant maps. In the case of the SPS30 model, we found that it presented an MNB of 0.44 and a CV of 2.76%, which assigned this model to Tier I. The possible uses of this sensor model are therefore related to the informal indication of pollutant presence.

### 6.2. EPA Guidelines Limits

The guidelines proposed by the EPA offer a practical tool that is useful for supporting the understanding of the most appropriate use, or application area, of an LCS or LCM, especially considering that the performance indicators related to the same sensor can present a wide range of values, as can be found in Table 6. The algorithm responsible for the “tier” selection proposed in the guidelines is based on the computation of the MNB and the CV. The first parameter takes into account the measurement discrepancies between the device under evaluation and the reference, while the CV provides a quantification of the consistency of the devices. Both these parameters are expressed by non-dimensional values, preserving the generality of the guidelines.

However, it can also be noted that the decision process for the selection of the different “tiers” lacks an indicator through which the grade of the correlation of the device with the reference is taken into account. This suggests that the addition of such an indicator to the mechanism at the basis of the tier selection could improve the effectiveness of the guidelines. The most suitable candidates could be the R^2^ and the slope of the linear fit line, because they are both non-dimensional parameters.

Nevertheless, a more detailed analysis of these indicators showed that the coefficient of determination (R^2^) depends on the range of reference measurements and the duration of the on-field evaluation. In particular, the wider the range of the pollutant concentration values registered by the reference device, the higher the probability of obtaining R^2^ values close to unity [11] is. This suggests that the selection of the slope of the linear fit line (see Figure 6) could be an additional indicator to use together with the MNB and the CV for “tiers” determination, but such a decision must be corroborated by targeted, detailed studies, which was beyond the limits of this work. Another limit of this research was the number of LCSs evaluated, the type of reference device considered, and the temporal frequency of the records comprising the measurement database (e.g., 5min, daily, or hourly averages). More specifically, this preliminary investigation performed an evaluation on the basis of daily averages but did not clarify if the application area (“tier”) of the evaluated sensors may be assigned to different “tiers” by changing the temporal average of the measurements (e.g., by considering hourly averages) or by changing the reference type.

## 7. Conclusions

The EPA guidelines offer a useful tool for assessing the most suitable application area, or use, of an LCS/LCM for air quality monitoring. To the best of our knowledge, until now, the performance of the PM sensor models considered in this work had not been analyzed in light of the EPA guidelines by other studies. Therefore, the SPS30 sensor produced by Sensirion and the PMS5003 sensor produced by Plantower were tested in this study using the SentinAir device as a LCM. Their performance in measuring PM_10_ concentrations was evaluated through an on-field test using a gravimetric reference providing daily averages of PM_10_.

We found that including the effects of the humidity by applying a correction algorithm proposed in previous works could improve the performance of the PMS5003 sensor models. We also found that, as prescribed by the guidelines, considering the MNB and CV indicators, the PMS5003 sensor could be used for supplementing regulatory monitoring networks, while the SPS30 sensor could be used to provide informal information about the presence of pollutants. However, even though the EPA guidelines represent without a doubt an efficient tool to guide the user through the jungle of the low-cost devices for air quality monitoring available on the market, we found that some improvements to the selection criteria characterizing the guidelines are necessary.

In our opinion, adding an indicator that takes into account the grade of correlation between the reference and the device under evaluation could make the guidelines more efficient in providing indications about the most appropriate application area. However, a rigorous analysis to confirm this aspect could represent a matter for future studies.

## Figures and Tables

**Figure 1 sensors-23-03976-f001:**
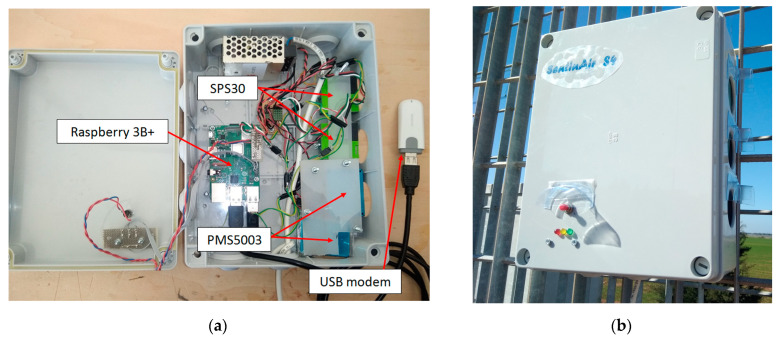
The SentinAir platform: (**a**) the device used in the experiment with the four PM sensors installed inside; (**b**) external view of the device acting as an LCM in this experiment.

**Figure 2 sensors-23-03976-f002:**
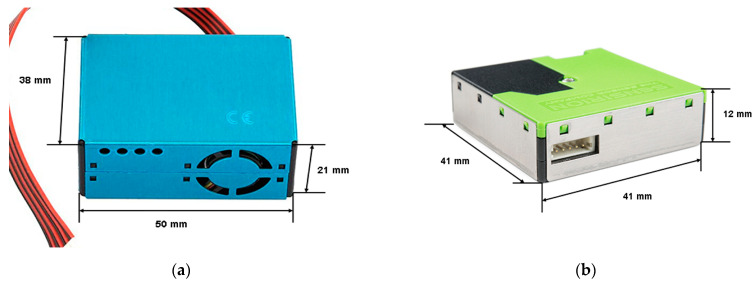
The miniaturized sensors used in this study: (**a**) the PMS5003 produced by Plantower and its size; (**b**) the SPS30 produced by Sensirion and its size.

**Figure 3 sensors-23-03976-f003:**
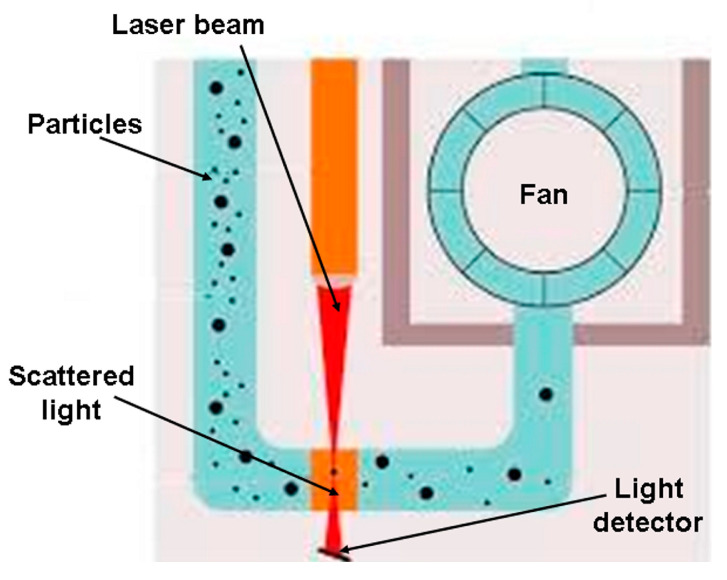
The simplified scheme of an optical counter for PM detection on which the working principle of LCSs used in this research is based.

**Figure 4 sensors-23-03976-f004:**
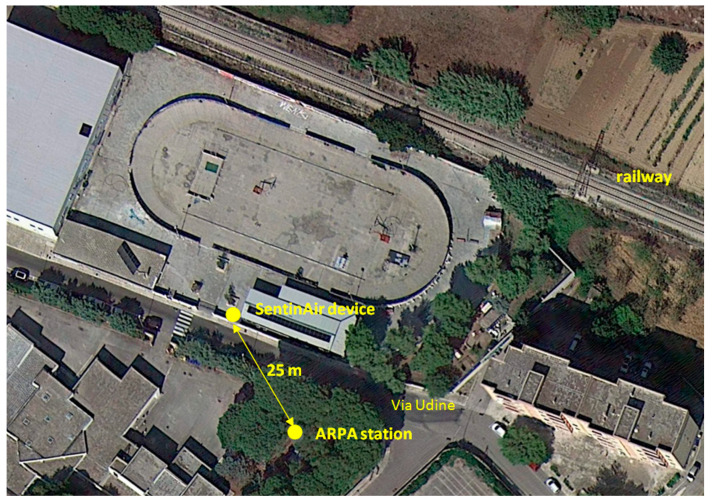
Depiction of the site where the ARPA station was located.

**Figure 5 sensors-23-03976-f005:**
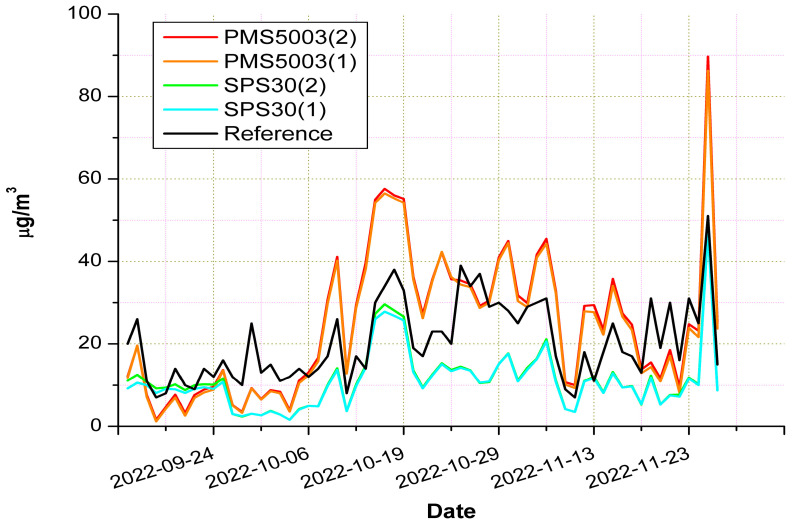
Time series of sensors under evaluation compared with the reference device.

**Figure 6 sensors-23-03976-f006:**
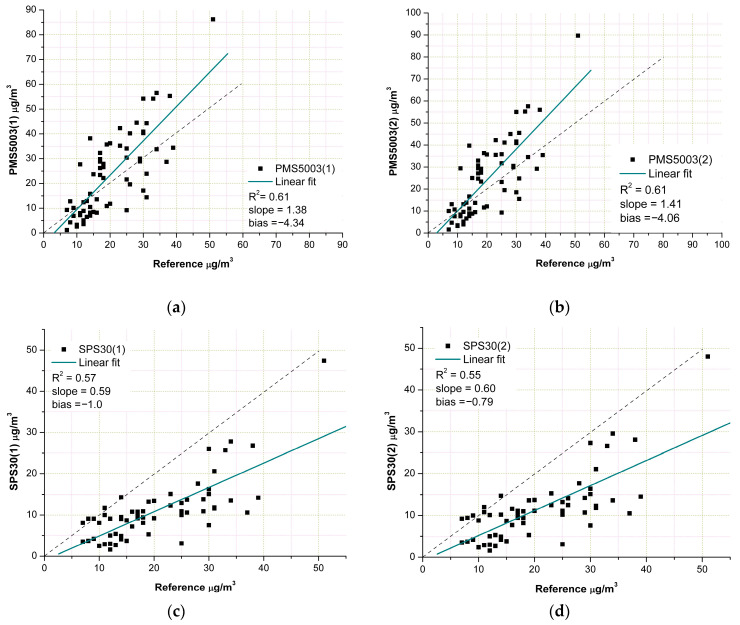
(**a**) PMS5003(1); (**b**) PMS5003(2); (**c**) SPS30(1); (**d**) SPS30(2). Comparison between the four copies of the PM sensors involved in this research and the reference device. The solid line represents a linear regression fit computed through the ordinary least squares method, while the dashed line indicates the 1:1 reference line. In the corners of the figures, the statistics are reported. The slope and bias are related to the slope and the intercept of the linear fit.

**Figure 7 sensors-23-03976-f007:**
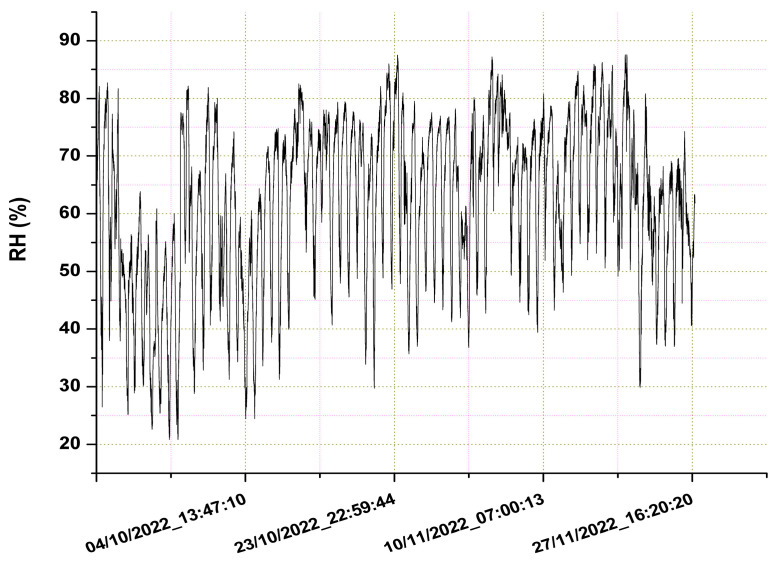
Time series of the relative humidity registered during the experiment.

**Figure 8 sensors-23-03976-f008:**
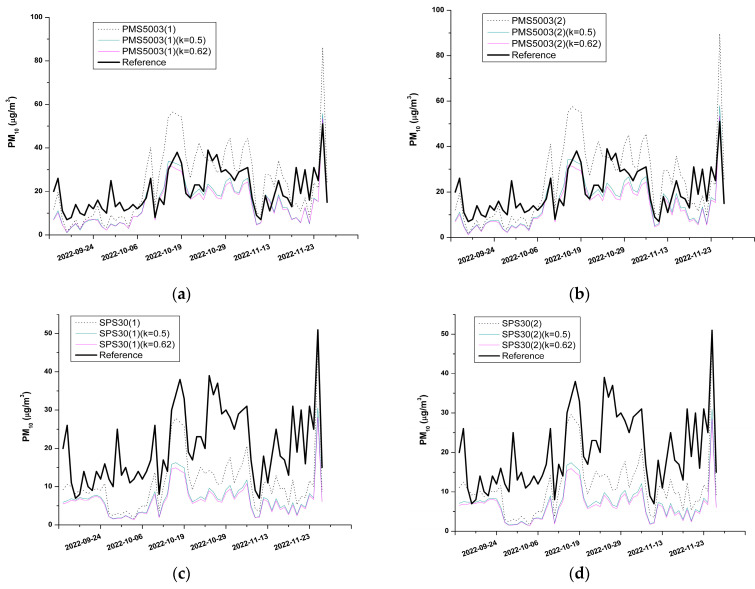
(**a**) PMS5003(1); (**b**) PMS5003(2); (**c**) SPS30(1); (**d**) SPS30(2). Time series of the sensor measurements compared with the reference device (thick black line). The dotted line is relative to the sensor data without the application of Equations (6) and (7). The straight thinner lines indicate the sensor measurements after applying the correction to include the humidity effect. The labels k = 0.5 and k = 0.62 indicate the different values of the “k” parameter in Equation (7).

**Table 1 sensors-23-03976-t001:** Classification of possible uses, or application areas, for LCSs/LCMs proposed in EPA guidelines.

Tier	Application Area	Pollutants	MNB	CV	Application Examples
I	Education and information	All	−0.5 < MNB < 0.5	CV < 0.5	Providing informal information about the presence of a pollutant; the use of sensors as teaching tools
II	Hotspot identification and characterization	All	−0.3 < MNB < 0.3	CV < 0.3	The identification of emission sources of pollutants such as heavy traffic or industrial facilities
III	Supplemental monitoring	O_3_, NO_2_, PM, CO, SO_2_, and TVOCs	−0.2 < MNB < 0.2	CV < 0.2	Supplementing the regulatory network monitoring for improving the spatio-temporal resolution of pollutant maps
IV	Personal exposure monitoring	All	−0.3 < MNB < 0.3	CV < 0.3	These sensors can be used in mobile monitors of a size that can be easily carried by users for measuring pollutant concentrations in indoor/outdoor environments
V	Regulatory monitoring	O_3_ CO, SO_2_, PM_10_, PM_2.5_, and NO_2_	−0.07 < MNB < 0.07; −0.1 < MNB < 0.1; −0.15 < MNB < 0.15	CV < 0.07 CV < 0.2 CV < 0.15	Pollutant monitoring to determine if an area complies with the national ambient air quality standards

**Table 2 sensors-23-03976-t002:** A summary of some LCMs/LCSs for PM measurements currently available on the market with their indicative costs and manufacturers.

Device Name	Cost (EUR)	Manufacturer	Device Type
PMS5003	~20	Plantower	LCS
SPS30	~50	Sensirion	LCS
OPC-N2	~350	Alphasense	LCS
SDS011	~33	Nova Fitness	LCS
PurpleAir PA-II	~190	PurpleAir	LCM
Airly PM	~900	Airly	LCM
Airquality Egg 2022	~630	Airquality Egg	LCM
TSI Bluesky	~380	TSI	LCM

**Table 3 sensors-23-03976-t003:** Coefficient of correlations related to the sensors involved in the experiment.

	PMS5003(2)	PMS5003(1)	SPS30(2)	SPS30(1)
**PMS5003(2)**	1.000	0.999	0.790	0.819
**PMS5003(1)**	0.999	1.000	0.785	0.813
**SPS30(2)**	0.790	0.785	1.000	0.997
**SPS30(1)**	0.819	0.813	0.997	1.000

**Table 4 sensors-23-03976-t004:** Performance indicators of each copy of the sensors in relation to the reference measurements. The MNB and CV values are in relation to the two sensor models considered in this research.

Sensor	R^2^	MAE	RMSE	Slope	Bias	MNB	CV
PMS5003(2)	0.61	9.56	12.09	1.41	−4.06	0.14	1.96%
PMS5003(1)	0.61	9.3	11.63	1.38	−4.34		
SPS30(2)	0.55	9.19	11.02	0.60	−0.79	−0.44	2.76%
SPS30(1)	0.57	9.47	11.26	0.59	−1.0		

**Table 5 sensors-23-03976-t005:** The performance indicators of the sensors under evaluation. The rows indicating PMS5003(1), PMS5003(2), and so on, show the data related to the uncorrected measurements, while the labels k = 0.5 and k = 0.62 indicate the data calculated by setting k = 0.5 and k = 0.62 in Equation (7). Data reported in bold characters denote the best performance.

Sensor	R^2^	MAE	RMSE	Slope	Bias
PMS5003(2)	0.61	9.56	12.09	1.41	−4.06
PMS5003(2) (k = 0.5)	**0.65**	**6.59**	**8.24**	**0.81**	**−11.48**
PMS5003(2) (k = 0.62)	0.65	7.21	8.84	0.74	−12.62
PMS5003(1)	0.61	9.3	11.63	1.38	−4.34
PMS5003(1) (k = 0.5)	0.65	6.81	8.5	0.75	−12.59
PMS5003(1) (k = 0.62)	0.65	7.21	8.84	0.81	−11.48
SPS30(2)	0.55	9.19	11.02	0.60	−0.79
SPS30(2) (k = 0.5)	0.50	13.31	15.03	1.42	−7.23
SPS30(2) (k = 0.62)	0.48	13.85	15.6	1.52	−6.81
SPS30(1)	**0.57**	**9.47**	**11.26**	**0.59**	**−1.0**
SPS30(1) (k = 0.5)	0.52	13.56	15.24	1.50	−6.72
SPS30(1) (k = 0.62)	0.51	14.1	15.8	1.62	−6.28

**Table 6 sensors-23-03976-t006:** A summary of the performance indicators determined in this work and previous studies related to LCMs using the SPS30 and PMS5003 sensors. Data refer to the PM_10_ concentration measurements. The ranges of the indicators express the minimum and maximum values found, considering that, in the same study, several copies of the same model were evaluated using different types of reference devices.

LCS Model	LCM Model	R^2^	RMSE	MAE	CV	Slope	Bias	Reference
PMS5003	SentinAir	0.61	11.63–12.09	9.3–9.56	1.96%	1.38–1.41	−4.06/−4.34	This study
PMS5003	Airly	0.27–0.47	20.5–21.4	16.4–17.8	1.3%	1.17–1.24	−23.4/−46.3	AQ-SPEC [18]
PMS5003	Airly	0.71–0.89	3.79–11.29	-	-	0.64–0.7	−0.11/−1.61	Castell [23]
PMS5003	Airquality Egg 2022	0.27–0.62	23.1–24.9	16.3–18.6	4.2%	1.01–1.72	−24.6/−29.14	AQ-SPEC [18]
PMS5003	PurpleAir PA-II	0.68–0.74	-	-	≅ 0%	1.21–1.70	−0.6/−20.2	AQ-SPEC [18]
PMS5003	Redspira	0.35–0.52	31.2–35.3	28–32.6	5.8%	1.02–1.37	−37.1/−41.4	AQ-SPEC [18]
PMS5003	Smart citizen kit II	0.10–0.17	-	-	6%	2.61–3.05	−157.0/−198.6	AQ-SPEC [18]
PMS5003	Lunar outpost	0.06–0.08	-	-	4.3%	2.63–3.38	−124.3/−140.9	AQ-SPEC [18]
SPS30	Ensense	0.9	1.92–2.26	-	z-	0.26–0.32	−9.39/−9.79	Castell [23]
SPS30	TSI Bluesky	0.28–0.34	-	-	11%	0.33–0.53	−4.5/−11.6	AQ-SPEC [18]
SPS30	Atmotube Pro	0.31–0.39	-	-	5.6%	0.83–0.98	−17.0/−31.4	AQ-SPEC [18]
SPS30	-	0.18–0.30	-	-	2.4%	0.77–1.61	−18.3/−19.9	AQ-SPEC [18]
SPS30	SentinAir	0.55–0.57	9.19–9.47	11.02–11.26	2.76%	0.59–0.60	−0.79/−1.0	This study

## Data Availability

Data are available by contacting the author at domenico.suriano@enea.it.

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
