# Peer review of "An Investigation on the Possible Application Areas of Low-Cost PM Sensors for Air Quality Monitoring"

_sensors, 2023, doi:10.3390/s23083976_

Round 1

Reviewer 1 Report

The article is well written, clear and useful. I do not have much punctual remarks except for:

·       Equation 3 MNB: Both summations use the index i

·       Equation 5, CV: It is not clear what measurement si,j, means. The measurement s has 2 indices. Explain what the indices mean.

My most important point of concern is that the novelty of the article – the LCS performance assessed by the EPA guidelines - should be emphasized more. I propose that the information that is mentioned in the introduction and the formulas in the experimental part is merged together in a section “2. Background”. That section should contain a description of (1) a description of the importance of standardized measurements (now it is mentioned that everybody is doing it in its own way but there is no clear standard suggested), (2) the information about Table 1 (Put the pollutants in a separate column as is the case for Table 5-1 from reference 19) together with the formulas, and (3) how the assessment is done (What happens when a sensor scores well for one indicator and bad for another indictor). In that way the evaluation flowchart becomes more explicit.

The discussion describes the weak points of the evaluation flowchart (data collected under different conditions, lack of indicator, etc.). I suggest that the evaluation of the “EPA assessment method” and its weak points is mentioned more explicitly.

Author Response

dear reviewer,

thank you for your suggestions aimed to improve the present manuscript. I have done my best to meet your requests, and here below the modifications are reported point by point:

        Equation 3 MNB: Both summations use the index i

  • Equation 5, CV: It is not clear what measurement si,j, means. The measurement s has 2 indices. Explain what the indices mean.

You are right, some mistakes were present in the formulas, now I have corrected them and added the necessary explanations (see lines 179-183).

 I propose that the information that is mentioned in the introduction and the formulas in the experimental part is merged together in a section “2. Background”. That section should contain a description of (1) a description of the importance of standardized measurements (now it is mentioned that everybody is doing it in its own way but there is no clear standard suggested), (2) the information about Table 1 (Put the pollutants in a separate column as is the case for Table 5-1 from reference 19) together with the formulas, and (3) how the assessment is done (What happens when a sensor scores well for one indicator and bad for another indictor). In that way the evaluation flowchart becomes more explicit.

A new section has been created in the body of the document accordingly with the given indications. In the new section (2. Background) it is stressed the absence of standardized measurements and the importance to set a minimum of rules to allow the performance comparison (see lines 154-167). It also contains the modified table 1 and how the assessment is done as requested (see lines 190-195).

The discussion describes the weak points of the evaluation flowchart (data collected under different conditions, lack of indicator, etc.). I suggest that the evaluation of the “EPA assessment method” and its weak points is mentioned more explicitly.

The discussion section is now split in two subsections. The second one highlights the weak points of EPA guidelines.

Reviewer 2 Report

Regarding format, the paper is well written. However, although the information provided in this paper could be useful for users of Low-cost sensors and devices, I think the paper needs to increase in research contributions. Maybe a proposal to improve the performance of low-cost sensors by mathematical models that allow obtaining a result closer to that of high-end sensors. Also, there are many self-citations for the amount of total references in the text. They should be reduced.

Furthermore, I recommend checking the following aspects:

-          I recommend adding a separated section for the related work right after the introduction. The text starting at line 83 up to line 140 seems to me to be more fitted in the related work section.

-          I also recommend extending the resulting Introduction to reach from 600 to 800 words. Also, at the end of the introduction, a paragraph could be added with the organization of the paper.

-          The use of the word “Anyway” in line 144 seems informal. I recommend changing it for synonyms such as “nonetheless”.

Author Response

Dear reviewer,

thank you for your suggestions. I made my best to meet your indications, and here below you can find point by point the answers to your requests.

Maybe a proposal to improve the performance of low-cost sensors by mathematical models that allow obtaining a result closer to that of high-end sensors

The study proposed in this paper is an investigation aimed to understand the potentialities of the sensors under the light of EPA guidelines as they are sold by their manufacturers. This type of study (at the best of our knowledge) is not yet addressed in previous works. In this manuscript, it is also reported a comparison with previous studies concerning the same sensor models. The calibration of the sensors and their performance improvement through data post-processing deserves a detailed work that cannot be exhaustively exposed in this manuscript for sake of conciseness, but that could be surely matter of  thorough future works.

 there are many self-citations for the amount of total references in the text. They should be reduced.

Accordingly to the given suggestion, I have decreased the number of reference to my co-authored previous works and increased the reference to other previous studies.

 I recommend adding a separated section for the related work right after the introduction. The text starting at line 83 up to line 140 seems to me to be more fitted in the related work section.

Done as requested

I also recommend extending the resulting Introduction to reach from 600 to 800 words. Also, at the end of the introduction, a paragraph could be added with the organization of the paper.

Done as requested

The use of the word “Anyway” in line 144 seems informal. I recommend changing it for synonyms such as “nonetheless”.

Done as requested

Reviewer 3 Report

sensors- 2278680

Title: An investigation on the possible application areas of low-cost PM sensors for air quality monitoring

Indeed, the manuscript is well-written and easy to follow. Some points need to be known.

-Please replace 'on' with 'of' in the title, e.g., An investigation of the possible application areas of low-cost PM sensors for air quality monitoring

-It will be good to add a reference for equations 1-5.

-During the investigation of PM sensors, have authors considered the atmospheric conditions like temperature, humidity etc.?

-Please add a table showing different PM sensors with cost and working principle.

-Please mark the parts name shown in Figure 1.

-How did the authors log the air quality data?

-How the authors have taken the results in Figure 5. Is there any experimental setup? Please clarify.

-The novelty of the work should be highlighted (in the Abstract and the conclusions).

Author Response

Dear reviewer,

thank you for your suggestions. I made my best to meet your indications, and here below you can find point by point the answers to your requests.

Please replace 'on' with 'of' in the title, e.g., An investigation of the possible application areas of low-cost PM sensors for air quality monitoring

Done as requested

It will be good to add a reference for equations 1-5

Done as requested (see line 173)

During the investigation of PM sensors, have authors considered the atmospheric conditions like temperature, humidity etc.?

Yes, environmental parameters such as humidity and temperature were logged, but as this research is focused in investigating the potentialities of the sensors as they are provided by the manufacturers under the light of EPA guidelines, they are not relevant for the study purposes, and therefore, not exposed.

Please add a table showing different PM sensors with cost and working principle

Done as requested (see table 2)

Please mark the parts name shown in Figure 1.

Done as requested

How did the authors log the air quality data?

Air quality data are stored in the Sd card memory present in the SentinAir device and also they can be downloaded thanks to the USB modem which allows a remote link. This aspect is now clarified in the manuscript by adding the following paragraph in the “Methods and materials” section (see lines 287-292): “The SentinAir system can store the data related to the performed measurements on its internal SD card memory; moreover, it is featured by a web server from which the user can download any data measured by the device thanks to the presence of a USB modem [35-37]. The data gathered by the SentinAir device were periodically downloaded and joined with the ARPA measurements publically available on its website [32].”

How the authors have taken the results in Figure 5. Is there any experimental setup? Please clarify.

By having clarified in the last point how the data were logged and how they were retrieved, now it is also clear that the results shown in figure 5 are the graphical representation of the database built with SentinAir data and ARPA measurements

The novelty of the work should be highlighted (in the Abstract and the conclusions).

I partially agree with the reviewer: in the abstract is clearly (but obviously in a concise way) stated that: “Until today, very few studies have analyzed the LCS performance by referring to the EPA guidelines. This research aims to understand the performance and the possible application areas of two models of PM sensors (the PMS5003, and the SPS30) on the basis of EPA guidelines.”. On the contrary, I have to admit that in the conclusion section this is not explicitly remarked. Therefore, to stress the novelty of this research, I added in the conclusion section (see lines 486-491) this element.

Round 2

Reviewer 1 Report

The article “An investigation on the possible application areas of low-cost PM sensors for air quality monitoring” intends to present possible application areas of two low-cost PM sensor models (PMS5003 and SPS30) for air quality monitoring. The manuscript is well written, more explicit, and the information is clearly presented. The remarks have been correctly corrected. I consider that the manuscript can be published.

Author Response

dear reviewer,

thank you for your precious support and suggestions for improving the manuscript.

Reviewer 2 Report

I think the research contributions of this paper are still falling short. If it would be to remain as a comparison of low cost sensors, then I think the comparison should be done with at least 10 of them. It should consider a good number of the most representative sensors in the market. Furthermore, there are still many self-citations.

Author Response

Dear reviewer,

Here below you can find my answers to your considerations, point by point:

  • I think the research contributions of this paper are still falling short. If it would be to remain as a comparison of low cost sensors, then I think the comparison should be done with at least 10 of them.  It should consider a good number of the most representative sensors in the market

There are countless works in which way less than 10 models of sensor, or 10 air quality monitors, or less than 10 sensors have been “just” evaluated. For example, we have: Marques, G., Ferreira, C.R. & Pitarma, R. Indoor Air Quality Assessment Using a CO2 Monitoring System Based on Internet of Things. J Med Syst 43, 67 (2019). https://doi.org/10.1007/s10916-019-1184-x, where a system based on a single CO2 sensor has been evaluated. This manuscript has been cited 71 times. Another example: Susanne Steinle, Stefan Reis, Clive E. Sabel, Sean Semple, Marsailidh M. Twigg, Christine F. Braban, Sarah R. Leeson, Mathew R. Heal, David Harrison, Chun Lin, Hao Wu, Personal exposure monitoring of PM2.5 in indoor and outdoor microenvironments, Science of The Total Environment, Volume 508, 2015, Pages 383-394, https://doi.org/10.1016/j.scitotenv.2014.12.003, cited 229 times, where JUST ONE model of PM2,5 monitor has been evaluated. Another example: Tingting Cao, Jonathan E. Thompson, Personal monitoring of ozone exposure: A fully portable device for under $150 USD cost, Sensors and Actuators B: Chemical, Volume 224, 2016, Pages 936-943, https://doi.org/10.1016/j.snb.2015.10.090. In this last manuscript, cited 26 times,  “just” one model of monitor “just” for ozone measurements was presented and evaluated. By sake of conciseness, I mean that the quality of a paper surely does not depend on the number of sensors involved in the experiment.

  • Furthermore, there are still many self-citations

The self-citations constitutes the 13% of the total number of citations.

Reviewer 3 Report

Sensors- 2278680

An investigation on the possible application areas of low-cost PM sensors for air quality monitoring

Thank you for allowing me to revise resubmitted manuscript titled " An investigation on the possible application areas of low-cost PM sensors for air quality monitoring" I believe the submitted manuscript and presented work is suitable for publishing in the Sensors, except for one minor revision.

Minor revision:

It will be good to add some more references related to PM sensors for air quality monitoring, e.g., Khan, A.U.; Khan, M.E.; Hasan, M.; Zakri, W.; Alhazmi, W.; Islam, T. An Efficient Wireless Sensor Network Based on the ESP-MESH Protocol for Indoor and Outdoor Air Quality Monitoring. Sustainability 2022, 14, 16630. https://doi.org/10.3390/su142416630

Author Response

Dear reviewer,

thank you for your suggestions and contribution, here below you can find my answers to your requests:

  • It will be good to add some more references related to PM sensors for air quality monitoring, e.g., Khan, A.U.; Khan, M.E.; Hasan, M.; Zakri, W.; Alhazmi, W.; Islam, T. An Efficient Wireless Sensor Network Based on the ESP-MESH Protocol for Indoor and Outdoor Air Quality Monitoring. Sustainability 2022, 14, 16630. https://doi.org/10.3390/su142416630

Done as requested

Round 3

Reviewer 2 Report

There are published papers on everything you want. They can be very cited as well. I understand that. But I still think that an effort could have been done to increase the research contribution. To add a proposal or something more to what you had already done.